# Main Concept, Sequencing, and Story Grammar Analyses of Cinderella Narratives in a Large Sample of Persons with Aphasia

**DOI:** 10.3390/brainsci11010110

**Published:** 2021-01-15

**Authors:** Jessica D. Richardson, Sarah Grace Dalton, Kathryn J. Greenslade, Adam Jacks, Katarina L. Haley, Janet Adams

**Affiliations:** 1Department of Speech and Hearing Sciences, University of New Mexico, Albuquerque, NM 87131, USA; jadams3@unm.edu; 2Department of Speech-Pathology and Audiology, Marquette University, Milwaukee, WI 53233, USA; sarahgrace.dalton@marquette.edu; 3Department of Communication Sciences and Disorders, University of New Hampshire, Durham, NH 03824, USA; Kathryn.Greenslade@unh.edu; 4Department of Allied Health Sciences, University of North Carolina at Chapel Hill, Chapel Hill, NC 27599, USA; adam_jacks@med.unc.edu (A.J.); Katarina_Haley@med.unc.edu (K.L.H.)

**Keywords:** aphasia, AphasiaBank, Cinderella, discourse, episode, main concept analysis, MSSG, narrative, sequencing, story grammar

## Abstract

Recently, a multilevel analytic approach called Main Concept, Sequencing, and Story Grammar (MSSG) was presented along with preliminary normative information. MSSG analyses leverage the strong psychometrics and rich procedural knowledge of both main concept analysis and story grammar component coding, complementing it with easy-to-obtain sequencing information for a rich understanding of discourse informativeness and macrostructure. This study is the next critical step for demonstrating the clinical usefulness of MSSG’s six variables (main concept composite, sequencing, main concept+sequencing, essential story grammar components, total episodic components, and episodic complexity) for persons with aphasia (PWAs). We present descriptive statistical information for MSSG variables for a large sample of PWAs and compare their performance to a large sample of persons not brain injured (PNBIs). We observed significant differences between PWAs and PNBIs for all MSSG variables. These differences occurred at the omnibus group level and for each aphasia subtype, even for PWAs with very mild impairment that is not detected with standardized aphasia assessment. Differences between PWAs and PNBIs were also practically significant, with medium to large effect sizes observed for nearly all aphasia subtypes and MSSG variables. This work deepens our understanding of discourse informativeness and macrostructure in PWAs and further develops an efficient tool for research and clinical use. Future research should investigate ways to expand MSSG analyses and to improve sensitivity and specificity.

## 1. Introduction

### 1.1. Main Concept Analysis

Main concept analysis (MCA) is a discourse analysis method that was introduced by Nicholas and Brookshire in 1993 [1,2] and has undergone further development since revitalization by Kong in 2009 [3] and the introduction of checklists and norms by Richardson and Dalton in 2015 [4]. The goal of MCA is to characterize and quantify the amount and accuracy of the essential elements, or gist, relayed by a speaker. Main concepts (MCs) are relevant utterances (containing one main verb, constituent nouns, and associated clauses) that are most commonly spoken by healthy controls about a shared discourse stimulus (e.g., picture, procedure, and story) and are thus deemed essential. To carry out MCA, discourse samples are coded for utterances that match the MCs in available checklists, and identified MCs are then scored for accuracy and completeness [1,2,3,4,5]. Following continual development and refinement, MCA has emerged as a strongly recommended approach because of its psychometric strengths and clinical feasibility [6]. Utilization of detailed scoring guidelines [2] and checklists (e.g., [4,5]) has resulted in consistent reports of high inter- and intra-rater reliability [1,2,3,4,5,7,8,9,10]. Additionally, test–retest reliability is adequate for repeated measurement [2,7,8,9] and certain MC codes (AC: accurate, complete; AI: accurate, incomplete; AB: absent) are stable enough to be recommended for use in group studies of discourse in aphasia [9].

MCA is sensitive to differences between control speakers and PWAs [2,3,8,11,12,13], as well as between persons with different profiles or subtypes of aphasia [8,12,13]. While the bulk of this work has focused on stroke-induced aphasia, recent application of MCA to a small sample (*n* = 17) of persons with primary progressive aphasia (PWPPAs) showed that they produced fewer accurate and complete (AC) MCs compared to controls [14]. With regard to both external criterion and ecological validity, MCA correlates with standardized measures of overall aphasia severity [7,8,13], confrontation naming (for some subtypes) [15], listener perceptions [16,17], conversational abilities [18], communication confidence [19], and life participation [19]. Combining MCA with temporal information for derived efficiency measures also seems promising, and preliminary norms have recently been introduced [14]. The number of accurate and complete (AC) MCs per minute (AC/min, or ACMC/min) is sensitive to differences between healthy controls and both PWAs [8,14] and PWPPAs [14].

### 1.2. Story Grammar Analysis

Despite the obvious strengths and utility of MCA, an informativeness-only focus ignores macrostructural elements necessary for successful communicative interactions, such as logical organization of discourse elements and adequate coverage of complex discourse topics. Story grammar measures can address these critical discourse variables in part by characterizing and quantifying story grammar elements (i.e., setting, initiating event, direct consequence, etc.) presented by PWAs [20,21]. Topic coverage can then be examined by determining how many episodes a PWA attempted to express, where episodic completeness is marked by the presence of the following: initiating event, attempt, and direct consequence [20,22] (but see Ulatowska et al. [23,24] for an alternative list of story grammar components that determine episodic completeness). If episodes are consistently produced with at least two of the three necessary components, then the speaker is expressing partially or completely complex episodes [10,25]. The addition of sequencing information (i.e., ordering of story grammar components/episodes presented from beginning to end) can potentially complement story grammar measures to provide information about organization and conceptualization [26,27].

Compared to MCA, much less is known about story grammar in PWAs. Most evidence is drawn from studies with small sample sizes that also utilize a wide range of theoretically linked but operationally diverse methods; therefore a discussion of original individual studies is warranted. For the purposes of this discussion, we focus on monologic semi-spontaneous spoken language tasks elicited via picture description and story retelling, excluding personal stories, procedures, etc. Early studies [23,24] examined story grammar (under titles of “discourse grammar”, “linguistic framework”, and “narrative superstructure”) and showed that PWAs with mild to moderate impairment were able to produce structured narrative discourse for varied genres. In a study of 10 more mildly impaired PWAs (7 mild, 3 mild–moderate) [23], all narratives contained the story grammar components required to meet criterion for complete episodes (i.e., setting, complicating event, and resolution for the study), though more variability was observed with optional components (e.g., codas, abstracts, evaluation). The sequencing of story grammar components was also preserved for all narratives according to subjective ratings. Authors summarized that overall, there were no significant differences between mildly impaired PWAs and controls for story grammar and sequencing variables. In a study of 15 moderately impaired PWAs [24], significant differences between PWAs and controls were reported, but their importance was somewhat diminished when the authors concluded that participants demonstrated preserved narrative structure. Compared to controls, moderately impaired PWAs had fewer optional and essential components (i.e., setting, resolution), and thus fewer overall complete episodes. In a longitudinal study throughout the first year of recovery for a person with mild fluent aphasia [22], despite gains in PICA scores (from 71st percentile to 93rd percentile) over time, story grammar did not improve—whereas controls generated 4–5 episodes per discourse task, the PWA never generated more than 2 episodes over 12 months of tracking. This discrepancy in recovery trajectory between overall communication abilities and story grammar was inconsistent with previous reports of mildly impaired PWAs and pointed to the need for the latter to be a focus for assessment and intervention.

Bottenberg and colleagues [28,29] examined “narrative level” according to Applebee’s [30] 6-level rating system for narrative structure and connections. The lowest rating is a 1 for “heaps”, or unlinked mentions of characters and/or actions; the highest rating is a 6 for “narrative” which contains a central theme, a forward momentum, and climax, and is more advanced than lower rated “simple sequences” and “focused chains”. Narrative level differed significantly between PWAs (9 mild, 1 moderate) and controls [28]. Bottenberg and colleagues [31] also applied a different 8-level rating system based on Stein and Glenn [20] and Hedberg and Stoel-Gammon [32] to study discourse in 12 mild–moderately impaired PWAs. The lowest rating is a 0 for “unrelated statements” and the highest rating is a 7 for “interactive episode”, which is more advanced than complex episodes and involves at least two characters whose goals and actions influence one another. While the study focus was to determine the impact of varied story elicitation stimuli on a wide range of discourse variables, authors noted that most participants produced complete stories with mean story grammar level ratings that ranged from 4 to 4.5 (4 = abbreviated episodes, 5 = completed episodes).

Whitworth [33,34] investigated story grammar outcomes (under the heading “coherence”) following a novel discourse treatment, *N*ovel *A*pproach to *R*eal-life communication: *N*arrative *I*ntervention in *A*phasia (NARNIA). In an early study [33], two PWAs (1 fluent, 1 non-fluent) were asked to tell the Cinderella story, and their production of “main events” were counted and organized by story grammar (i.e., setting, actions, resolution). While no further details were provided about this analysis (e.g., what constitutes a main event?), Whitworth demonstrated with this measure that story grammar significantly improved for both participants following NARNIA. In a later study [34] of 14 PWAs (7 mild, 7 moderate), more details regarding coding of utterances (C-units, as in Miller and Iglesias [35]) and story grammar components for different discourse genres (recount, procedure, exposition, and narrative) were provided. Focusing on the narrative (Cinderella), authors coded C-units as one of the following categories: (1) setting/orientation; (2) body, with subcodes for initiating event, response/plan, event, evaluative comment; and (3) conclusion. PWAs in both NARNIA and standard care groups demonstrated numerical improvements in narrative story grammar for the three categories, with the largest effect size in the body category for the NARNIA group, but only statistically significant improvement in the body category for the standard care group. This counterintuitive finding may be related to the variance (i.e., *SD*) within the groups. More notable changes in story grammar were observed with the other discourse genres assessed.

### 1.3. Linking Main Concept and Story Grammar Analyses

Story grammar measures have recently been lauded for their psychometric strengths [36] and there is interest in combining two such strong approaches—MCA and story grammar [13]. Indeed, promising endeavors have been explored and will be briefly discussed. Early work by Ulatowska et al. [23,24] “determined pragmatically” [23] (p. 360) the essential steps and propositions for procedures and narratives (conceptually similar to MC checklists) in the same study where they investigated story grammar. There was no further information regarding list development, the lists themselves, or coding procedures, and related quantitative results and/or qualitative impressions were inconsistently presented. Though not synergistically utilized, this was an early example of side-by-side investigation of MCA-like and story grammar analyses, revealing that while PWAs were able to maintain overall structure, they included fewer essential steps or propositions, as well as fewer essential and/or optional story grammar components, and their performance depended on aphasia severity.

In a four-year longitudinal study of a single participant with non-fluent aphasia (global → Broca’s), Stark [37] investigated recovery of discourse content and structure using Cinderella retellings. To do so, she collected discourse samples from a small sample of healthy controls to identify a list of 41 possible content units or propositions, with 23 propositions further categorized as essential or constituent (analogous to MC checklists). She further organized these propositions into setting/orientation, episodes (5 total), and coda. Coding for accuracy and completeness, as in MCA, was not included. With this approach, she characterized recovery of narrative content and related narrative structure. While the participant was able to produce at least some propositions related to each story grammar division (e.g., episode 1, episode 2) throughout the monitoring period, improvements in quality (telegraphic propositions → more complete propositions) and quantity (more propositions overall) were observed over time.

Lê and colleagues [25,38] introduced the Story Goodness Index (SGI), a two-dimensional classification system of discourse organization and completeness. SGI first involves calculation of a story grammar score, where utterances (T-units, as in Hunt [39]) coded as one of three main story grammar components required to constitute an episode (i.e., initiating event, attempt, or direct consequence), but not other components (e.g., setting), are totaled. Next, the utterances are compared to a list of essential utterances (analogous to MC checklists) so that a story completeness score is calculated by counting utterances that were correctly produced (similar to an accurate and complete MC, or AC MC [13]). Story grammar or organization (*y* axis) is then plotted against story completeness (*x* axis), and 1- and 2-standard deviation (*SD*) cutoffs are then used to create quadrants for interpretation, resulting in what the authors coined SGI. Quadrant 1 is low content-high organization, 2 is high content-high organization, 3 is low content-low organization, and 4 is high-content-low organization. To date, SGI has been applied to the study of cognitive and linguistic outcomes following penetrating and closed traumatic brain injury, revealing differences for SGI and/or its constituent measures between persons with brain injury and controls (e.g., [25,38,40,41,42,43]) and characterizing recovery patterns [44].

### 1.4. Sequencing

Still needed are measures that go beyond counting of components to consider the sequencing of those components. It is entirely possible for persons with linguistic and/or cognitive deficits to produce narratives with essential concepts that match up to essential story grammar components, but to present them out of sequence and/or with revisions and backtracking, which would negatively impact communication. Hameister and Nickels [27] combined a modified MCA with sequencing information (i.e., order of MCs) to analyze picture scene description in 50 controls and 50 PWAs (mild to severe, varied subtypes). They generated MC checklists for their task, and since they were interested in conceptualization more than linguistic expression, spoken attempts were given credit (i.e., any MC code except for absent [AB]) along with non-verbal attempts (e.g., pointing to pictured elements, relationships, and/or actions). During MC checklist development, the authors determined the median MC order (i.e., which MC was most commonly presented first, second, etc.) and introduced a Difference-in-Order ratio (DiO ratio) to express the difference between the expected versus produced order of MCs. This ratio is the total number of actual order differences divided by the total number of possible order differences; a higher ratio corresponds to more out-of-order MCs (see appendix C of Hameister and Nickels [27] for calculation matrix). Even with the generous MC coding approach described above, PWAs produced significantly fewer MCs than controls, while DiO ratios were more variable. Upon closer examination, half of the sample demonstrated notable reduction in MCs whereas only 9 PWAs differed significantly for DiO compared to controls.

### 1.5. Main Concept, Sequencing, and Story Grammar Analyses

As is clear from this brief review, MCA is well established, and story grammar has been analyzed with a variety of methods. It would be advantageous to combine these approaches and to operationalize them for research and clinical utilization. To that end, we recently introduced a multilevel analytic approach called Main Concept, Sequencing, and Story Grammar (MSSG) along with preliminary normative information [10]. MSSG leverages the strong psychometrics and rich procedural knowledge of both MCA and story grammar component coding, complementing it with easy-to-obtain sequencing information. To begin, the 34 MCs for Cinderella established by Richardson and Dalton [4] were pre-assigned one of six story grammar component codes (or code options in the case of five MCs): setting, initiating event, attempt, direct consequence, mental state (i.e., internal response or reaction), and conclusion [20,21]. MCs were also pre-assigned an episode number (1–5) (or episode options in the case of 2 MCs). To initiate scoring, utterances that matched MCs were transferred from orthographic transcripts (along with line numbers) into the scoring worksheet. MCs were coded for accuracy and completeness according to established rules and guidelines [2,4], assigned a numeric value (0–3) representing the continuum of accuracy and completeness [3,4], and totaled, to yield the *main concept (MC) composite* score. Each MC then received a *sequencing* score, using sequencing rules [10] to assign to each MC a numeric value (0–3) representing order, and totaled. These two scores were then summed for the *main concept + sequencing* (*MC + Sequencing*) score. For the five episodes, the presence (1) or absence (0) of each of the three required episodic story grammar components—initiating event, attempt, and direct consequence—was marked and then tallied for the *total episodic components* score. Finally, episodes with at least two of those required components were assigned 1 point as a complex episode (0 points if 0–1 components) and totaled across episodes for the *episodic complexity* score.

Using the same control sample from Richardson and Dalton [4], Greenslade et al. [10] reported preliminary norms for typical adults without brain injury with stratification by age bins (i.e., 20–39, 40–59, 60–79, and ≥80 years). Results were consistent with previous MCA findings [4] and with expectations for a typically aging sample—most participants produced narratives that were loaded high for content, episodic complexity, and organization, with better performance for the younger age groups compared to the older age groups. Inter-rater reliability for all story grammar and sequencing variables was “very good”, with Cohen’s kappa ranging from 0.99 to 1.0 and point-to-point agreement ranging from 99 to 100%. These findings were a first important step towards demonstrating the utility of this multilevel analytic approach [6] in that it is grounded in relevant theory, measures specific constructs, and, with the tools provided [2,4,10], facilitates scoring that is reliable across raters. Further, with an eye towards avoiding barriers to both research and clinical utilization [14], MSSG involves a brief elicitation task, simple orthographic transcription, relatively straightforward coding (following training and with use of instructional materials), objective quantitative scores, and preliminary control norms for comparison.

### 1.6. Study Purpose

This study is the next critical step for demonstrating the clinical usefulness of Cinderella MSSG for PWAs. We present descriptive statistical information for MSSG variables for a large sample of PWAs and compare their performance to a large sample of persons without a reported history of brain injury or disease. Compared to controls not brain injured, we expect PWAs to perform significantly differently for all MSSG variables, though magnitude of difference is expected to vary according to subtype. Importantly, we also expect to see that MSSG is sensitive to differential performance of even the mostly mildly impaired PWAs, as in previous research [11,12,13,14]. Finally, we expect different performance profiles, but with overlapping values, for different aphasia subtypes.

## 2. Materials and Methods

### 2.1. Participants

Transcripts from 370 persons with aphasia (PWAs) and 112 persons not brain injured (PNBIs) were retrieved from the AphasiaBank database and from the laboratories of authors (as they are being finalized for submission to AphasiaBank). The contributing labs and investigators include the following (with the AphasiaBank file identifier in parentheses): Adler Aphasia Center (Adler), Aphasia Center of California (Elman), Aphasia Center of Tucson (Tucson), Aphasia Center of West Texas (ACWT), Aphasia House (Whiteside), Aphasia Lab of the University of South Carolina (Fridriksson), Boston University (BU), Carnegie Mellon University (CMU), East Carolina University (Wright), Emerson College (Kempler), InteRACT: Intensive Residential Aphasia Communication Therapy (Wozniak), Montclair State University (MSU), Northwestern University (Thompson), private practice (Garrett), Stroke Comeback Center (Williamson), Snyder Center for Aphasia Life Enhancement (SCALE), Stroke Aphasia Recovery Program (STAR), Texas Christian University (TCU), Triangle Aphasia Project (TAP), University of Kansas (Kansas), University of Kentucky (Capilouto), University of Massachusetts-Amherst (Kurland), University of New Hampshire (UNH), University of New Mexico (Richardson), and University of North Carolina-Chapel Hill (Haley, Jacks).

See Table 1 for complete demographics by group and aphasia subtype. Most WAB-R [45] aphasia subtypes were initially represented in the data. Subtype groups with fewer than 10 individuals in the sample (i.e., global and transcortical sensory) were only included in the omnibus group comparisons and regression; they were excluded for the pairwise group comparisons. Also included were individuals who had a stroke and reported functional aphasia but who scored above the WAB-R cutoff, referred to in this paper (as in previous papers (e.g., [11,12,13,14,15,16])) as “not aphasic by WAB” (NABW). Based on WAB-R classification, aphasia subtypes included the following: 122 anomic, 85 Broca’s, 67 conduction, 4 global, 54 NABW, 12 transcortical motor, 2 transcortical sensory, and 24 Wernicke’s. The average WAB-R Aphasia Quotient (AQ) score was 72.8 (*SD* = 19.8). For PWAs, the average age was 62 years (*SD* = 12.6), with an average education of 15.3 years (*SD* = 2.9). A subset of PWAs (238/370) in this study have been included in a previous normative MCA study by Dalton and Richardson [13].

Two of the 112 PNBIs were removed from the data as extreme outliers who did not contribute meaningful variance to the data. In a clinical scenario, their communication abilities would not be classified as “normal” despite the absence of a medical diagnosis of brain injury or neurodegenerative disease and would warrant assessment. The average age of PNBIs was 58.3 years (*SD* = 20.8) with an average education of 15.8 years (*SD* = 2.5). The majority of PNBIs (95/110) have been included in previous normative MCA and MSSG studies [4,5,10,13].)

### 2.2. Transcripts

All discourse samples were elicited using the AphasiaBank protocol and procedures (https://aphasia.talkbank.org/protocol/). Participants reviewed the picture book by Grimes [46] with the words covered prior to attempting the storytelling. They were then instructed: “Now tell me as much of the story of Cinderella as you can. You can use any details you know about the story, as well as the pictures you just looked at”. For AphasiaBank CHAT (Codes for Human Analysis of Transcripts) transcripts, the Cinderella story retell was extracted to be coded and scored for the variables described below. For transcripts awaiting submission, CHAT transcripts or orthographic transcripts were utilized. Line numbers were added to all extracted transcripts.

### 2.3. Scoring

#### 2.3.1. Main Concepts

Transcripts were scored for MCs using standardized checklists [4] and scoring procedures [2]. Each MC consisted of 2–4 numbered essential elements (e.g., ^1^ Prince ^2^ falls in love ^3^ with Cinderella). Occasionally, non-essential elements that commonly co-occurred with the essential elements were included to contextualize the main concept (e.g., [The king thinks] ^1^ the prince ^2^ should get married). Checklists include alternative productions (not exhaustive) to aid in scoring (e.g., alternatives for “falls in love” include “is enamored with”, “is delighted with”, “is awestruck by”, “likes”, and “is hooked on”). According to Nicholas and Brookshire [1,2], an MC consists of one main verb and its constituent arguments, as well as prepositional phrases and/or subordinate clauses that operate on the main verb, as appropriate. Each participant’s transcript was scored for the presence or absence of MCs. Missing MCs were coded as absent (AB); MCs that were present could be assigned one of four codes based upon the accuracy and completeness of the essential elements within each MC. An accurate/complete (AC) code was assigned if all essential elements were present and correct. An accurate/incomplete (AI) code was assigned if one or more essential elements were missing but all essential information that was produced was correct. An inaccurate/complete (IC) code was assigned if all essential elements were present but some essential elements were inaccurate based on control speakers’ productions. Finally, an inaccurate/incomplete (II) code was assigned if one or more essential elements were missing and one or more of the essential elements that were produced were inaccurate. MC codes were transformed to numeric scores using the formula adapted from Kong [3] with a maximum score of 102 (34 MCs multiplied by 3 points):(#AC × 3) + (#AI × 2) + (#IC × 2) + (#II × 1) = *MC composite* score(1)

#### 2.3.2. Sequencing

In order to determine whether MCs were produced in a logical sequence, line numbers were added to the orthographic transcripts. To be logically sequenced, each MC had to come after the MC(s) before it (e.g., MC 1 followed by MC 2 followed by MC 3), with exceptions for when the sequence was judged to be interchangeable. For example, MCs 24–26 (“^1^ The prince ^2^ finds ^3^ Cinderella’s shoe”, “^1^ Everything ^2^ turns back ^3^ to its original form”, and “^1^ She ^2^ returned ^3^ home”) could be presented in any sequence among themselves. Sequencing exceptions are outlined in appendix A of Greenslade et al. [10]. MCs stated in a logical sequence were assigned 3 points. MCs stated out of sequence but signaled as being in the wrong sequence by the speaker (e.g., “I forgot to say”) were assigned 2 points. MCs stated out of sequence without signaling or revising were assigned 1 point. Absent MCs were assigned a sequencing score of 0. If all 34 MCs were present and logically sequenced, the maximum *sequencing* score was 102 (34 correctly sequenced MCs multiplied by 3 points).

#### 2.3.3. Main Concept + Sequencing

A combined *main concept + sequencing* (*MC + Sequencing*) score was calculated by adding the MC and sequencing scores for each concept. An utterance corresponding to a main concept could receive a maximum score of 6 for this measure if it was correctly sequenced (3 points) and if it received an MC code of accurate/complete (3 points). MC+Sequencing scores were then summed across the 34 main concepts for a maximum total of 204 (34 accurate and complete MCs multiplied by 3 points, plus 34 recognizable MC attempts in correct order multiplied by 3 points).

#### 2.3.4. Essential Story Grammar Components

Twenty-nine of the 34 MCs were pre-assigned one of six story grammar component codes: setting, initiating event, attempt, direct consequence, mental state (i.e., internal response or reaction), and conclusion [20,21]. The remaining 5 MCs were assigned coding options since the order of their production, neighboring MCs, or specific formulation determined their story grammar component. For example, the story grammar component assigned to MC 17 “^1^ Cinderella ^2^ went to ^3^ the ball” was determined based on whether the verb indicated Cinderella was in transit to the ball (e.g., “^2^ left for ^3^ the ball”, “^2^ goes to ^3^ the ball”) and was therefore coded as a direct consequence; and/or whether the verb indicated Cinderella was at the ball (e.g., “^2^ arrives at ^3^ the ball”, “^2^ reaches ^3^ the ball”) and therefore was coded as an initiating event. Each MC that received a code other than AB (absent) also received 1 point for the corresponding story grammar component. The one exception to this rule is MC 17, where 2 points could be received if the individual indicated both that Cinderella was going to the ball and that she arrived at the ball. The maximum *essential story grammar components* score was 35, if all 34 MCs were present and MC 17 included both meanings. This variable was not reported in the original MSSG study [10] but is included here as it is an informative count of recognizable attempts at essential concepts and story grammar components and can be compared to previously reported MC Attempts [11,13].

#### 2.3.5. Total Episodic Components

MCs were also pre-assigned an episode number (1–5), or episode options in the case of MC 17 “^1^ Cinderella ^2^ went ^3^ to the ball” and MC 18 “^1^ She ^2^ had to be ^3^ home by midnight”, where order and formulation determined the story grammar component and corresponding episode. For the five episodes, the presence (1) or absence (0) of each of the three required episodic story grammar components-initiating event, attempt, and direct consequence-is marked and then tallied for the *total episodic components* score. If each required episodic story grammar component is produced for each episode, the maximum score is 15 (5 episodes × 3 episodic story grammar components).

#### 2.3.6. Episodic Complexity

Complex episodes are defined as episodes produced with at least two required episodic story grammar components (i.e., initiating event, attempt, direct consequence). Each complex episode received 1 point while episodes with no or only 1 episodic story grammar component produced received 0 points. Points were totaled across episodes for the *episodic complexity* score (maximum score of 5 if all episodes were complex).

### 2.4. Fidelity and Reliability

#### 2.4.1. Assessment Fidelity

A majority of the transcripts were collected and transcribed by many different AphasiaBank contributors (see Section 2.1. Participants), potentially introducing variability to data collection. We have previously examined this for the Cinderella story retelling [4], finding excellent adherence to the AphasiaBank protocol instructions for administration and excellent transcription reliability (less than 1% of total content words were omitted, added, or incorrect). The remaining transcripts (awaiting submission to AphasiaBank) were orthographically transcribed by a research assistant and then checked for accuracy by another research assistant who updated the file as needed.

The MC checklist [4] and training manual have been updated and refined over years of development, and we have been able to determine for which MCs or scoring rules raters encounter the most difficulty. For example, commonly visited rules from Nicholas and Brookshire [2] involve accuracy and completeness decisions about pronoun referents, and we find that different interpretation can lead some raters to score an utterance as AI (accurate, incomplete) and others to score the same utterance as IC (inaccurate, complete). While this does not impact the MC composite score because both AI and IC codes are worth 2 points, it is important for ensuring fidelity and for future work with this dataset. We also have discovered inconsistent application of the completeness rule “Statements containing some of the essential information”, which allows for non-MC utterances to be partially used if it establishes an important story component (e.g., character and pivotal event), and so have expanded the manual with examples. Finally, we have further expanded the manual with transcript examples encountered when making consensus decisions. Main concepts that were previously scored [13] were reviewed and updated as needed. New transcripts were scored with these updated resources following rater training as described below.

The first three authors (Richardson, Dalton, and Greenslade) performed scoring as well as training duties for this study. Graduate students completed approximately 2–4 weeks of training for assigning main concepts, sequencing, and/or story grammar scores. Training duration depended on which variables they were assigned (some or all) and on their training performance. Training was completed when raters reached at least 80% reliability for point-to-point agreement with pre-scored transcripts generated by study authors. Debrief discussions (via email, phone, videoconferencing) were held to discuss scoring differences to maximize learning from errors.

#### 2.4.2. Inter-Rater Reliability

Point-to-point agreement [47] (or rather concept-by-concept or component-by-component) inter-rater reliability was calculated for 20% of the study sample (22 PNBIs, 76 PWAs) to ensure fidelity and reliability. For PNBIs, inter-rater reliability was as follows: 96.8% for MC composite, 98% for sequencing, 97% for essential story grammar components, 99% for total episodic components, and 100% for episodic complexity. For PWAs, inter-rater reliability was as follows: 95.4% for MC composite, 98.7% for sequencing, 97.9% for essential story grammar components, 95.3% for total episodic components, and 97.6% episodic complexity.

### 2.5. Data Analysis

#### 2.5.1. Normative Data

In order to maximize clinical utility of these measures, we report descriptive statistics (mean, median, *SD*, range, skew, and kurtosis) for all PNBIs, all PWAs, and for each aphasia subtype. Normality was evaluated using skew and kurtosis where skew >±2 and kurtosis >±4 indicated a non-normal distribution [48,49]. Homogeneity of variance was visually evaluated using histograms and revealed that PWAs generally had positively skewed distributions (with the bulk of values clustering around low scores) while PNBIs generally had negatively skewed distributions (with the bulk of values clustering around higher scores).

#### 2.5.2. Comparisons between PNBIs and PWAs

As in Dalton and Richardson [13], omnibus and pairwise group comparisons were conducted to identify differences between PNBIs and PWAs, as well as between PNBIs and the aphasia subtypes. Since assumptions of normality and homogeneity of variance were violated for all variables, median tests were used to evaluate between-group differences. Effect sizes (phi; φ = √[χ^2^ ÷ N]) are also reported to identify comparisons that are more likely to reflect practically, rather than statistically, significant differences [50]. Similar to the interpretation of other effect sizes, 0.1 ≤ φ < 0.3 indicates a small effect, 0.3 ≤ φ < 0.5 indicates a medium effect, and φ ≥ 0.5 indicates a large effect [51].

To further support diagnostic utility and characterize group differences, the percent overlap between PNBIs and each aphasia subtype is reported (PNBI—overlap—PWA subtype), as in Dalton and Richardson [13] who modified the analysis from McNeil et al. [52]. First we identified the highest value for each MSSG variable within each aphasia subtype (PWA-high). Then we counted how many PNBIs scored at or below that value (# of PNBIs ≤ PWA-high). Finally, we divided the number of PNBIs scoring at or below that value by the total number of PNBIs and multiplied by 100 to obtain a percentage ([# of PNBIs ≤ PWA-high] ÷ [total PNBIs]) × 100.
([# of PNBIs ≤ PWA-high] ÷ [total PNBIs]) × 100 = *percent PNBI overlap*(2)

For example, when comparing the PNBI distribution to the anomic distribution, the highest MC composite score for anomics was 75. Of the 110 total PNBIs, 89 received an MC composite score of seventy-five or less, which means that 81% of PNBIs’ scores fall within the range, or overlap, with the scores of PWAs with anomic aphasia (i.e., [89 ÷ 110] × 100 = 81%). The percent overlap of PNBIs with each aphasia subtype distribution provides a rough estimate of the specificity of the measure for diagnostic purposes, with greater overlap indicating poorer specificity (or reduced ability to obtain a true negative).

A similar calculation determining the percent overlap of each aphasia subtype distribution with the PNBI distribution for each MSSG variable was also completed (PWA subtype—overlap—PNBI). First we identified the lowest value for each MSSG variable within the PNBI group (PNBI-low). Then we counted how many of each aphasia subtype scored at or above that value (# of PWAs ≥ PNBI-low). Finally, we divided the number of PWAs (per subtype) scoring at or above that value by the total number of PWAs (per subtype) and multiplied by 100 to obtain a percentage ([# of PWAs ≥ PNBI-low] ÷ [total PWAs] × 100).
([# of PWAs ≥ PNBI-low] ÷ [total PWAs]) × 100 = *percent PWA overlap*(3)

For example, when comparing the anomic distribution to the PNBI distribution, the lowest MC composite score for PNBIs was 15. Of the 122 anomics in the sample, 86 received an MC composite score of 15 or higher, indicating that 70% of anomic’s MC composite scores overlap with PNBIs MC composite scores (i.e., [86 ÷ 122] × 100 = 70%). The percent overlap of the PWA subtype distributions with the PNBI distribution provides a rough estimate of the sensitivity of the measure for diagnostic purposes, with greater overlap indicating poorer sensitivity (or reduced ability to obtain a true positive).

#### 2.5.3. Secondary Analysis

A secondary analysis of select demographic variables was conducted to fill a gap in our understanding of the influence of age, education, and gender on discourse outcomes. For healthy aging adults, there is ample evidence of changes in discourse due to typical aging (e.g., [4,5,10,53]). Additionally, higher level of educational attainment may predict better language abilities and better preservation of those abilities throughout typical adult development. Gender may also influence story retelling, especially for the Cinderella story (but see Fromm et al. [54]). How these demographic variables interact with brain injury such as stroke and subsequent outcomes is largely unknown. To explore these important variables, generalized linear model analyses were conducted. The goodness of fit of various underlying distributions was evaluated and compared across models using the Pearson chi-square statistic divided by degrees of freedom (Pearson χ^2^/*df*), log likelihood, and Akaike Information Criteria (AIC). Models with Pearson χ^2^/*df* values close to 1 indicate good fit to the distribution of the residuals, while log likelihood and AIC values closer to 0 indicate models that are well fit without being unnecessarily complex [55,56].

## 3. Results

### 3.1. Comparisons between PNBIs and PWAs

#### 3.1.1. Main Concept Composite

Descriptive statistics show that PWAs had lower scores, with a more restricted range compared to PNBIs (Table 2 and Figure 1a). All omnibus (χ^2^ = 227.6; *p* < 0.001; φ = 0.689) and pairwise comparisons (*anomic*: χ^2^ = 102.4, *p* < 0.001, φ = 0.664; *Broca’s*: χ^2^ = 146.1, *p* < 0.001, φ = 0.866; *conduction*: χ^2^ = 94.2, *p* < 0.001, φ = 0.729; *NABW*: χ^2^ = 27.1, *p* < 0.001, φ = 0.407; *transcortical motor*: χ^2^ = 12.5, *p* < 0.001, φ = 0.320; *Wernicke’s*: χ^2^ = 29.2, *p* < 0.001, φ = 0.467) were statistically significant. Large effects were observed between PNBIs and PWAs with anomic, Broca’s, and conduction subtypes. Medium effects were observed between PNBIs and PWAs with NABW, transcortical motor, and Wernicke’s subtypes. When examining the distributions, percent PNBI overlap with the aphasia subtypes was highly variable and was ordered from greatest distinction between groups (least overlap) to least distinction between groups (most overlap) as follows: *Wernicke’s*: 5%; *transcortical motor*: 8%; *Broca’*s: 13%; *conduction*: 39%; *anomic*: 81%; and *NABW*: 96%. When examining the distribution of percent PWA overlap with PNBIs, similar variability was observed, and percentages were ordered from greatest to least distinction as follows: *Broca’s*: 14%; *transcortical motor*: 25%; *Wernicke’s*: 33%; *conduction*: 56%; *anomic*: 70%; *NABW*: 99%. (See Table 3).

#### 3.1.2. Sequencing

Descriptive statistics show that PWAs had lower scores, with a more restricted range compared to PNBIs (See Table 2 and Figure 1b). All omnibus (χ^2^ = 199.6; *p* < 0.001; φ = 0.645) and pairwise comparisons (*anomic*: χ^2^ = 75.1, *p* < 0.001, φ = 0.569; *Broca’s*: χ^2^ = 139.2, *p* < 0.001, φ = 0.845; *conduction*: χ^2^ = 71.1, *p* < 0.001, φ = 0.634; *NABW*: χ^2^ = 20.7, *p* < 0.001, φ = 0.355; *transcortical motor*: χ^2^ = 12.9, *p* < 0.001, φ = 0.325; *Wernicke’s*: χ^2^ = 28.4, *p* < 0.001, φ = 0.460) were statistically significant. Large effects were observed between PNBIs and PWAs with anomic, Broca’s, and conduction subtypes. Medium effects were observed between PNBIs and PWAs with NABW, transcortical motor, and Wernicke’s subtypes. When examining the distributions, percent PNBI overlap with the subtypes was highly variable and was ordered from greatest to least distinction between groups as follows: *transcortical motor*: 14%; *Wernicke’s*: 25%; *Broca’*s: 33%; *conduction*: 64%; *anomic*: 87%; and *NABW*: 98%. When examining the distribution of percent PWA overlap with PNBIs, similar variability was observed, and percentages were ordered from greatest to least distinction as follows: *Broca’s*: 31%; *transcortical motor*: 33%; *Wernicke’s*: 46%; *conduction*: 67%; *anomic*: 76%; *NABW*: 98%. (See Table 3).

#### 3.1.3. Main Concept+Sequencing

Descriptive statistics show that PWAs had lower scores, with a more restricted range compared to PNBIs. All omnibus (χ^2^ = 211.5; *p* < 0.001; φ = 0.664) and pairwise comparisons (*anomic*: χ^2^ = 89.6, *p* < 0.001, φ = 0.622; *Broca’s*: χ^2^ = 142.2, *p* < 0.001, φ = 0.854; *conduction*: χ^2^ = 88.3, *p* < 0.001, φ = 0.706; *NABW*: χ^2^ = 24.8, *p* < 0.001, φ = 0.389; *transcortical motor*: χ^2^ = 12.5, *p* < 0.001, φ = 0.320; *Wernicke’s*: χ^2^ = 29.2, *p* < 0.001, φ = 0.467) were statistically significant. Large effects were observed between PNBIs and PWAs with anomic, Broca’s, and conduction subtypes. Medium effects were observed between PNBIs and PWAs with NABW, transcortical motor, and Wernicke’s subtypes. (See Figure 1c and Table 2). Percent PNBI overlap results were as follows: *transcortical motor*: 9%; *Wernicke’s*: 13%; *Broca’*s: 20%; *conduction*: 52%; *anomic*: 85%; and *NABW*: 96%. Percent PWA overlap results were as follows: *Broca’s*: 25%; *transcortical motor*: 33%; *Wernicke’s*: 38%; *conduction*: 63%; *anomic*: 74%; *NABW*: 98%. (See Table 3).

#### 3.1.4. Essential Story Grammar Components

Descriptive statistics show that PWAs had lower scores, with a more restricted range compared to PNBIs. All omnibus (χ^2^ = 205.8; *p* < 0.001; φ = 0.655) and pairwise comparisons (*anomic*: χ^2^ = 72.4, *p* < 0.001, φ = 0.559; *Broca’s*: χ^2^ = 130.7, *p* < 0.001, φ = 0.819; *conduction*: χ^2^ = 65.5, *p* < 0.001, φ = 0.608; *NABW*: χ^2^ = 19.7, *p* < 0.001, φ = 0.346; *transcortical motor*: χ^2^ = 12.9, *p* < 0.001, φ = 0.325; *Wernicke’s*: χ^2^ = 29.2, *p* < 0.001, φ = 0.467) were statistically significant. Large effects were observed between PNBIs and PWAs with anomic, Broca’s, and conduction subtypes. Medium effects were observed between PNBIs and PWAs with NABW, transcortical motor, and Wernicke’s subtypes (see Figure 1d and Table 2). Percent PNBI overlap results were as follows: *transcortical motor*: 16%; *Wernicke’s*: 25%; *Broca’*s: 34%; *conduction*: 70%; *anomic*: 96%; and *NABW*: 96%. Percent PWA overlap results were as follows: *Broca’s*: 27%; *transcortical motor*: 33%; *Wernicke’s*: 42%; *conduction*: 67%; *anomic*: 74%; *NABW*: 98%. (See Table 3).

#### 3.1.5. Total Episodic Components

Descriptive statistics show that PWAs had lower scores, with a more restricted range compared to PNBIs. All omnibus (χ^2^ = 194.5; *p* < 0.001; φ = 0.637) and pairwise comparisons (*anomic*: χ^2^ = 74.7, *p* < 0.001, φ = 0.568; *Broca’s*: χ^2^ = 130.7, *p* < 0.001, φ = 0.819; *conduction*: χ^2^ = 42.2, *p* < 0.001, φ = 0.488; *NABW*: χ^2^ = 20.7, *p* < 0.001, φ = 0.355; *transcortical motor*: χ^2^ = 8.6, *p* < 0.003, φ = 0.266; *Wernicke’s*: χ^2^ = 28.4, *p* < 0.001, φ = 0.460) were statistically significant. Large effects were observed between PNBIs and PWAs with anomic and Broca’s subtypes. Medium effects were observed between PNBIs and PWAs with conduction, NABW, and Wernicke’s subtypes. A small effect was observed between PNBIs and PWAs with transcortical motor aphasia. (See Figure 1e and Table 2). Percent PNBI overlap results were as follows: *transcortical motor*: 24%; *Wernicke’s*: 40%; *Broca’*s: 56%; *conduction*: 78%; *anomic*: 95%; and *NABW*: 95%. Percent PWA overlap results were as follows: *transcortical motor*: 25%; *Broca’s*: 26%; *Wernicke’s*: 42%; *conduction*: 64%; *anomic*: 74%; *NABW*: 98%. (See Table 3).

#### 3.1.6. Episodic Complexity

Descriptive statistics show that PWAs had lower scores, with a more restricted range compared to PNBIs. Omnibus (χ^2^ = 181.7; *p* < 0.001; φ = 0.615) and pairwise comparisons (*anomic*: χ^2^ = 44.8, *p* < 0.001, φ = 0.439; *Broca’s*: χ^2^ = 120.1, *p* < 0.001, φ = 0.785; *conduction*: χ^2^ = 19.8, *p* < 0.001, φ = 0.334; *NABW*: χ^2^ = 12.4, *p* < 0.001, φ = 0.274; *transcortical motor*: χ^2^ = 7.5, *p* < 0.006, φ = 0.248; *Wernicke’s*: χ^2^ = 14.3, *p* < 0.001, φ = 0.327) were statistically significant. A large effect was observed between PNBIs and PWAs with Broca’s aphasia. Medium effects were observed between PNBIs and PWAs with anomic, conduction, and Wernicke’s aphasia. Small effects were observed between PNBIs and NABW, and transcortical motor subtypes. (See Figure 1f and Table 2). Percent PNBI overlap results were as follows: *Broca’*s: 60%; *transcortical motor*: 60%; *Wernicke’s*: 60%; *anomic*: 100%; *conduction*: 100%; and *NABW*: 100%. Percent PWA overlap results were as follows: *Broca’s*: 34%; *Wernicke’s*: 46%; *transcortical motor*: 50%; *conduction*: 69%; *anomic*: 76%; *NABW*: 96%. (See Table 3).

#### 3.1.7. MSSG Classification

Inspired by the Story Goodness Index (SGI) [40,41], we plotted participants’ MC+Sequencing dimension (*x* axis) against their total episodic components dimension (*y* axis) for the MSSG Classification (see Figure 2). In doing so, it is easy to visualize the relationship between participants’ ability to tell accurate, complete, and logically sequenced stories (MC+Sequencing) and their ability to maintain overall episodic structure (total episodic components) [10]. Similar to SGI, quadrants were defined by cutoff points at 1 *SD* (solid lines) and 2 *SD* (dotted lines) from the control mean of both MSSG variables. (See Table 2 for *M* and *SD* values for each variable.) For MC+Sequencing, 1 *SD* below the mean was 89.3, and 2 *SD* was 55.1. For total episodic components, 1 *SD* below the mean was 9.2, and 2 *SD* was 6.8. We report here classification for Quadrant 2 (high sequenced content—high episodic structure) and Quadrant 3 (low sequenced content—low episodic structure). Using 1-*SD* values, the distribution of scores for Quadrant 2 (high scores) was as follows: *PNBI*: 81.8%; *NABW*: 38.9%; *anomic*: 22.1%; *conduction*: 11.9%; *transcortical motor*: 0%; *Wernicke’s*: 0%; *Broca’s*: 1.2%. Using 2-*SD* values, the distribution of scores for Quadrant 2 (high scores) was as follows: *PNBI*: 95.5%; *NABW*: 83.3%; *anomic*: 49.1%; *conduction*: 31.3%; *transcortical motor*: 16.7%; *Wernicke’s*: 25%; *Broca’s*: 7.1%. Using 1-*SD* values, the distribution of scores for Quadrant 3 (low scores) was as follows: *PNBI*: 14.5%; *NABW*: 42.6%; *anomic*: 64.8%; *conduction*: 76.1%; *transcortical motor*: 91.6%; *Wernicke’s*: 91.7%; *Broca’s*: 98.8%. Using 2-*SD* values, the distribution of scores for Quadrant 3 (low scores) was as follows: *PNBI*: 3.6%; *NABW*: 14.8%; *anomic*: 41.8%; *conduction*: 53.7%; *transcortical motor*: 75%; *Wernicke’s*: 66.7%; *Broca’s*: 90.5%. Though not pictured, the two PWAs-*transcortical sensory* and the four PWAs-*global* were all 100% in Quadrant 3 (low scores) for both 1-*SD* and 2-*SD* values.

### 3.2. Secondary Analysis

Given the statistically significant differences described above, it is clear that PNBIs and PWAs belong to separate populations, and that aphasia impacts discourse performance. For these reasons, generalized linear modelling was conducted separately on the two groups. For PNBIs, we compared models with normal, Poisson, and Tweedie distributions for all MSSG variables except episodic complexity, where we used the multinomial distribution given the small range in this variable (0–5) in place of the Tweedie. For PWAs, we compared models with normal, gamma, Poisson, and Tweedie distributions for all MSSG variables except episodic complexity, where again we included a comparison with a multinomial distribution. See Appendix A for goodness-of-fit statistics for all distributions tested for both groups.

Both age and gender were significant predictors of PNBIs’ discourse performance for all MSSG variables except total episodic components, where only age was a significant predictor. In contrast, education and gender were significant predictors of PWAs’ discourse performance for all MSSG variables. See Table 4 for full results of the significant predictors.

## 4. Discussion

To our knowledge, this study included the largest and most diverse sample of PWAs to examine main concepts, sequencing, and/or story grammar. We found significant differences between PNBIs and PWAs for all MSSG variables. These differences occurred at the omnibus group level and for each aphasia subtype, even for PWAs with very mild impairment that is not detected with standardized aphasia assessment (i.e., PWAs-NABW). Differences between PNBIs and PWAs were also practically significant, with medium to large effect sizes observed for nearly all aphasia subtypes and MSSG variables. The only exceptions were small effect sizes for PWAs-NABW for episodic complexity and for PWAs-transcortical motor for total episodic components and episodic complexity. Large effect size differences were commonly observed for anomic, Broca’s, and conduction aphasia subtypes, whereas medium effect size differences were usually observed for NABW, transcortical motor, and Wernicke’s subtypes. Given the amount of overlap between distributions for PNBIs and PWAs-NABW, a medium effect size (or small for episodic complexity) probably reflects the “true” magnitude of differences between the two samples. However, the small or medium (instead of large) effect sizes observed for the smaller groups (i.e., transcortical motor and Wernicke’s subtypes) should be interpreted with caution, as the effect size calculated for this study is strongly influenced by sample size.

Aphasia subtypes that consistently demonstrated the least overlap with PNBIs were Broca’s, transcortical motor, and Wernicke’s; these groups alternated rankings of first, second, and third lowest percentage overlap across variables. Conduction aphasia consistently ranked fourth, followed by anomic and then NABW aphasia. Since the aphasia subtypes can be approximately ordered based on overall severity, MC-related rankings were not surprising, as previous research highlighted moderate-to-strong correlations between WAB-R AQ (i.e., overall aphasia severity) and two MC scores, MC composite and MC attempts (where attempts are nearly identical to essential story grammar components) [13]. This study is the first to show the relationship between WAB-R subtype (and severity range estimate) information and story grammar and sequencing variables. Further examination of percent overlap data show that the most discriminatory MSSG variable seemed to be MC composite followed closely by MC+Sequencing. The least discriminatory was episodic complexity, and the other variables (i.e., sequencing, essential story grammar components, and total episodic components) fell in between. The MSSG classification plots further illustrate the combined discriminatory power of discourse content and organization in identifying distributions of narrative macrostructure performance across groups and the extent of challenges experienced by persons with each aphasia subtype.

Here we replicated and extended previous MCA findings [13] in a larger sample of PWAs than investigated previously. While story grammar findings have been less consistent, due in large part to variability (e.g., story grammar measures, participants) and small sample sizes, we demonstrated on a large scale with a wide range of aphasia subtypes and severity that narrative macrostructure is not as well preserved as previously believed and that it should be considered for assessment and treatment. Alongside previously developed norms [4,10], this study’s normative and comparative findings further demonstrate the utility of the multilevel MSSG analyses for complex narrative production using the Cinderella story retelling. Further, MSSG builds upon well-established and well-defined MCA procedures in a straightforward manner, yielding valuable, objective, and quantitative metrics of discourse performance across content and organization domains. Indeed, the most time-consuming aspect of MSSG analyses is the MCA, after which the remaining story grammar and sequencing scoring can be conducted with relative ease. Finally, like MCA, MSSG procedures also hold the potential for non-transcription based utilization. Taken together, it is clear that several key barriers to clinical implementation of discourse assessment have been addressed with MSSG development.

### 4.1. Spotlight on PWAs-NABW

There is an appalling lack of mainstream assessment measures sensitive to PWAs-NABW, who have also been described in other literature as having subclinical [57] or latent [58,59] aphasia. This means that these individuals rarely qualify for clinical services and also that both research and clinical treatment programming options for addressing higher-level deficits are scarce. In order to provide opportunities and services for this population, detailed characterization of their deficits is needed. We have previously demonstrated that PWAs-NABW differ significantly from PNBIs for main concepts (MCA) [11,12,13,14], core lexicon (CoreLex; a measure of typical lexical retrieval for shared discourse tasks) [11,12], and other microlinguistic measures such as number of utterances, lexical diversity (as measured via moving average type-token ratio [MATTR]), lexical entropy (as measured via word information measure [WIM]), and words per minute (WPM) [12,60]. We have also shown that this group performs significantly differently for the derived efficiency measure AC/min (or ACMC/min; accurate and complete main concepts produced per minute) but not for CoreLex/min (core lexical items produced per minute) [14]. PWAs-NABW perform most similarly to high-scoring PWAs-Anomic [11,12,13,14] but do differ significantly for some microlinguistic variables (e.g., WPM, MLU, MATTR [12]). Here we found that PWAs-NABW differed significantly from PNBIs, both statistically and practically, for MSSG variables. While percent overlap data were somewhat discouraging (consistently >95%), these significant findings coupled with the MSSG Classification visualization illustrate that while some PWAs-NABW may be difficult to distinguish from PNBIs, enough of the sample differs with a great enough magnitude to warrant investigation. Of particular interest is the notable change in percentages for PWAs-NABW when examining Quadrants 2 and 3 (Figure 2). Moving from 1-*SD* to 2-*SD* cutoffs for Quadrant 2 (high scores) involved an increase in over 40 percentage points; at 1-*SD*, only 38.9% of PWAs-NABW were considered to have logically sequenced, informative narratives with overall episodic structure, yet at 2-*SD*, 83.3% were then classified as good storytellers. A reverse but similar situation was observed for Quadrant 3. A large portion of this group resides in that no man’s land between 1- and 2-*SD* cutoffs, and it is critical to acknowledge these bearings and avoid setting arbitrary cutoffs for diagnostic decisions until we increase our understanding of the deficits and needs of PWAs-NABW. These findings alongside previous research demonstrate clearly that they are a unique group situated in between PWAs-Anomic and PNBIs and are in need of attention and intervention [19,60,61]. Taken together, this information has hopefully laid sufficient groundwork for a sea-change for aphasia diagnosis and treatment.

Further strengthening this standpoint, DeDe and Salis [58] recently developed and applied an interesting and novel analysis to examine the Cinderella story retelling in PWAs-NABW (referred to by authors as “latent aphasia”). They investigated temporal speech production variables (e.g., pauses, mazes, speech rate, repetitions, and revisions) and discourse organization variables (e.g., percentage of formulation time for utterances that continue episodes, episode omission, and episode recurrence) in controls, PWAs-NABW, and PWAs-Anomic. PWAs-NABW consistently differed from controls for roughly half of the variables under study, with statistically significant differences reported for the following: word count, silent pause duration, speech rate, and percentage of formulation time for utterances that continued episodes as well as utterances that introduced new episodes. The only significant differences between PWAs-NABW and PWAs-Anomic occurred for articulation rate, percentage of formulation time for utterances that continued episodes, and episode omission. PWAs-NABW produced narratives that did not differ significantly in episode or story completeness compared to controls but that were significantly reduced in content and speed. Authors highlighted three areas of deficit that could be contributing to these findings in PWAs-NABW—language planning, lexical retrieval, and processing speed—and emphasized processing speed as a critical factor to consider. They advocated for the inclusion of temporal information to discourse assessment to increase sensitivity. Additional work in this vein with persons with very mild deficits is likely to better characterize the widely suspected but difficult to unmask cognitive processing deficits in PWAs of varying subtypes and severities, which would eventually lead to improved treatment approaches and outcomes.

### 4.2. Limitations and Future Directions

#### 4.2.1. The Timing May be the Thing

In light of these recent findings regarding temporal differences in the Cinderella narrative [58], and how such measures may be particularly sensitive to latent aphasia, it is worthwhile to pursue temporal additions to current MSSG variables. A clinically feasible approach would be the addition of a derived efficiency measure, as we have utilized recently with main concept (MCA) and core lexicon (CoreLex) analyses [14]. Derived efficiency measures have proven to be reliable, sensitive, and ecologically valid [14] and we plan to publish normative information for these variables in the near future. An efficiency measure would require very little additional work on behalf of researchers or clinicians—simply timing the narrative and performing a simple calculation—but would likely be quite informative. The addition of the more rigorous protocol developed by DeDe and Salis [58] (or similar) would add even greater power, and it may be possible to move forward with one of the powerful measures from their study—silent pauses—via automated approaches (e.g., [62,63]).

#### 4.2.2. Errors and Macrostructure and Timing (Again), Oh My

Investigation of main concept (MC) codes was not included in this study, but their relationship to the other MSSG variables (e.g., sequencing, story grammar) should be explored. Recall that MCs, if present, can receive one of four scores: AC (accurate and complete); AI (accurate and incomplete); IC (inaccurate and complete); and II (inaccurate and incomplete) [2]. While previous research has shown that presence or absence of accurate and complete MCs (i.e., AC or AB [absent] codes) corresponds to overall language abilities as measured by WAB-R, the error codes (AI, IC, II) do not, perhaps indexing other aspects of language processing [13]. Andreetta and Marini [64] have reported on the deleterious impact of lexical retrieval difficulties on narrative cohesion and coherence in 20 persons with mild to severe fluent aphasia. They suggested that language difficulty experienced during connected speech can impact lexical selection for subsequent utterances as they unfold over time, leading to errors of cohesion and coherence. Hameister and Nickels [27] linked their sample’s more selective difficulty with verbs over nouns to poorer main concept production and suggested that conceptualization deficits may be important to consider. DeDe and Salis [58] also proposed several underlying areas of deficit related to poor narrative abilities, including the aforementioned lexical retrieval and conceptualization, but they focused on processing speed deficits in their study of latent aphasia. Processing speed may underlie those lexical retrieval and planning deficits and may be detectable via the temporal measures they studied. They also predicted that as narratives unfold and as processing demands increase, the presence of imprecise and/or errored lexical selection (and other errors) becomes more likely. The study of MC error codes, especially over time, and their relation to story grammar and sequencing variables certainly warrants further investigation. This would be complemented well with the recently developed word information measure (WIM), a measure of the novelty (or unexpected-ness) of words within word strings or utterances [60].

#### 4.2.3. Non-Essential Workers

An advantage of MCA and its expansion MSSG is that it is somewhat streamlined, requiring consideration only of the 34 MCs for Cinderella [4]. As we have stated in previous work [10,13], while this feature certainly assists with clinical utility, it does leave a lot of potentially rich information “on the table” and unexamined. Hameister and Nickels [27] investigated non-MCs directly, tallying utterances that were either marginally relevant (but not “main” or essential concepts) or served as commentary. Interestingly, PWAs produced more non-MCs, and the presence and quantity of non-MCs seemed to be positively correlated to their sequencing variable, the Difference-in-Order (DiO) ratio. Authors examined more closely their individual participant data and concluded that these findings only applied to a subset of their sample. They advocated for the monitoring of sequencing and non-MCs as a valuable predictor of conceptualization deficits that may involve difficulty prioritizing information to be communicated (i.e., foreground v. background). Processing speed deficits proposed by DeDe and Salis [58] may also relate to the inclusion of non-essential information. Specifically, as demands increase and/or as time elapses, diminishing access to relevant information may lead PWAs to abandon, fill, and/or substitute content in an attempt to meet perceived task demands. There is not likely a single cause for this phenomenon of added non- or less-relevant information, given the heterogeneity of lesion anatomy and subsequent aphasic deficits in PWAs; rather it is more likely that it is highly individualized and due to lexical retrieval deficits [64], planning and conceptualization deficits [27], and/or processing speed deficits [58], depending on lesion location and the health of the intact brain. Until a better understanding is gained, it is important to consider the nature of the stimuli as well as the task instructions when tracking measures related to relevant or irrelevant content, episodic structure, and/or temporal measures [27,58]. The presence or absence of perceived time pressure, presence or absence of instructions regarding how much detail to provide, and other factors can influence these variables as well as the inferences that researchers and clinicians can draw.

#### 4.2.4. Discourse Research and Diversity

In a secondary analysis, we explored the relationship between demographic variables (i.e., age, educational attainment, and gender) because of their known influence in typical aging and the general lack of knowledge with regard to how these variables interact with brain injury and language recovery. Age is believed to negatively correlate with stroke recovery (e.g., [65,66]) and there is evidence that females experience poorer recovery than their male counterparts (e.g., [67,68]). Educational attainment may positively correlate with stroke recovery (e.g., [66,69]), though some studies disagree (e.g., [65]. We showed that gender was a significant predictor of MSSG variables for both PNBIs and PWAs. However, age predicted additional variance for PNBIs performance, but not for PWAs, and educational attainment predicted additional variance for PWAs performance, but not for PNBIs. The finding that age did not predict PWAs performance was somewhat surprising—since our previous norms development work with PNBIs showed clear age effects (when age was sorted by bins 20–39, 40–59, 60–79, 80+) for both MCA and MSSG analyses [4,10], we expected to see age as a significant predictor. However, it appears that the presence of aphasia erases those age effects. In previous studies, we have recommended the further development and use of age-stratified norms, but it certainly seems that we need to consider the development of multifactor norms moving forward. This will require continued leveraging and expansion of valuable databases such as AphasiaBank to continue this important work. Moreover, it will require targeted expansion, as the current database, while large, has a somewhat restricted range for educational attainment that is not representative of many of the patients and participants we reach. Expansion will also need to focus on recruitment of more culturally and linguistically diverse speakers. Note that we did not address race/ethnicity for this study, for even though we increased the diversity of this sample with new data collection, AphasiaBank and our added laboratory samples are still primarily comprised of Caucasian speakers.

#### 4.2.5. Happily Ever … after Cinderella

The Cinderella story retelling is widely used for discourse assessment and therefore information regarding performance for a wide range of discourse variables is available for reference [70]. It is easy to administer, is familiar to several generations, yields more complex and lengthy narratives compared to other semi-spontaneous tasks, and with recently developed tools [4,10,71], administration and scoring can be relatively efficient. However, there are concerns regarding its ecological validity and its use in more culturally and linguistically diverse populations, and further development and investigation of other discourse genres is needed. Whitworth et al. [34] separated analysis of the Cinderella narrative from other discourse genres (i.e., recount, procedure, exposition), referring to the latter as “everyday discourse”. They found that improvement following a novel conversational treatment approach, NARNIA, was observed more for everyday discourse than for the Cinderella retelling. Whitworth et al. [34] (see also Pritchard et al. [36]) introduced a detailed protocol that may help guide expansion of MSSG to other discourse genres.

However, to analyze “everyday discourse” genres, different methods may be required. For example, high point analysis rather than story grammar analysis is typically used with personal narratives, as the expected organizational components may differ, including opening and closing appendages (i.e., abstract, coda), orientation, complicating action, resolution, and evaluation [72]. This alternate analysis approach accounts for differences in storytelling across genres. For example, commentary (e.g., “Guess what?”, “you won’t believe what happened next!”) is expected in personal narratives, whereas it may detract from fictional narrative, as identified by Hameister and Nickels [27]. Further because of the variable content conveyed in personal narratives, generating a list of main concepts is challenging, if not impossible, and clinical feasibility and replicability is thus drastically reduced. On the other hand, procedural discourse has been amenable to main concept analysis [4], and assessing sequencing would be a logical extension in some cases (e.g., Peanut Butter and Jelly Sandwich task). At the same time, no episodic structure is expected in procedural discourse, and thus, a different approach would be needed to assess organization, such as cohesion analysis and/or ratings (e.g., cohesion rating in the Expository Scoring Scheme [73]).

While personal narratives and procedural discourse may be more ecologically valid given their relevance in day-to-day interactions, it would be worth exploring whether MSSG analyses for fictional narratives, like Cinderella, may correspond with or predict variables from personal narrative or procedural discourse tasks. Further, it would be beneficial to examine which discourse analysis variables may best predict everyday conversational abilities, as reported by PWA and/or caregivers. Here again is an opportunity to leverage the AphasiaBank (or similar) database, as the AphasiaBank protocol also includes free speech samples (i.e., Stroke Story and Coping, Important Event).

## 5. Conclusions

Main concept analysis (MCA) is a replicable, psychometrically strong, and clinician-friendly approach that provides information about how accurately and completely a speaker conveys essential information during discourse. Story grammar analysis approaches have been used to determine how well a speaker is able to produce the structured “cognitive skeleton” of a story, and while there is a wide range of varied methodologies in use, they are trusted approaches. We recently developed a multilevel analytic approach that combined MCA and story grammar analysis and complemented it with a measure of sequencing. This approach, Main Concept, Sequencing, and Story Grammar (MSSG), addresses a critical gap in discourse assessment, in that there is consensus that psychometrically sound, operationally detailed, and clinician-friendly tools are needed for measuring functional communication as a primary outcome in aphasia. The following tools are required to carry out MSSG analyses and interpretation: rules for MC scoring in Nicholas and Brookshire [2]; MC checklists in Richardson and Dalton [4]; rules for scoring sequencing and story grammar in Greenslade et al. [10]; and normative and comparative information for PNBIs and PWAs in Richardson and Dalton [4], Dalton and Richardson [13], Greenslade et al. [10], and this current study. Our current findings demonstrated clearly that MSSG analyses were sensitive to differences between PNBIs and PWAs. Differences, both statistically and practically significant, were presented by subtype. Guidance for clinical interpretation of our findings was provided in the form of explanation and presentation of effect sizes, percent overlap data, and a MSSG Categorization that allows readers to visualize how PWAs included in this study perform relative to PNBIs. Readers can also use the normative information provided for the MSSG Categorization with their own participants and patients to compare their performance with the groups studied here, identify discourse strengths and weaknesses, develop subsequent treatment targets, and chart treatment responsiveness. This work deepens our understanding of discourse informativeness and macrostructure in PWAs and further develops an efficient tool for research and clinical use. Future research should investigate ways to expand MSSG analyses and to improve sensitivity and specificity.

## Figures and Tables

**Figure 1 brainsci-11-00110-f001:**
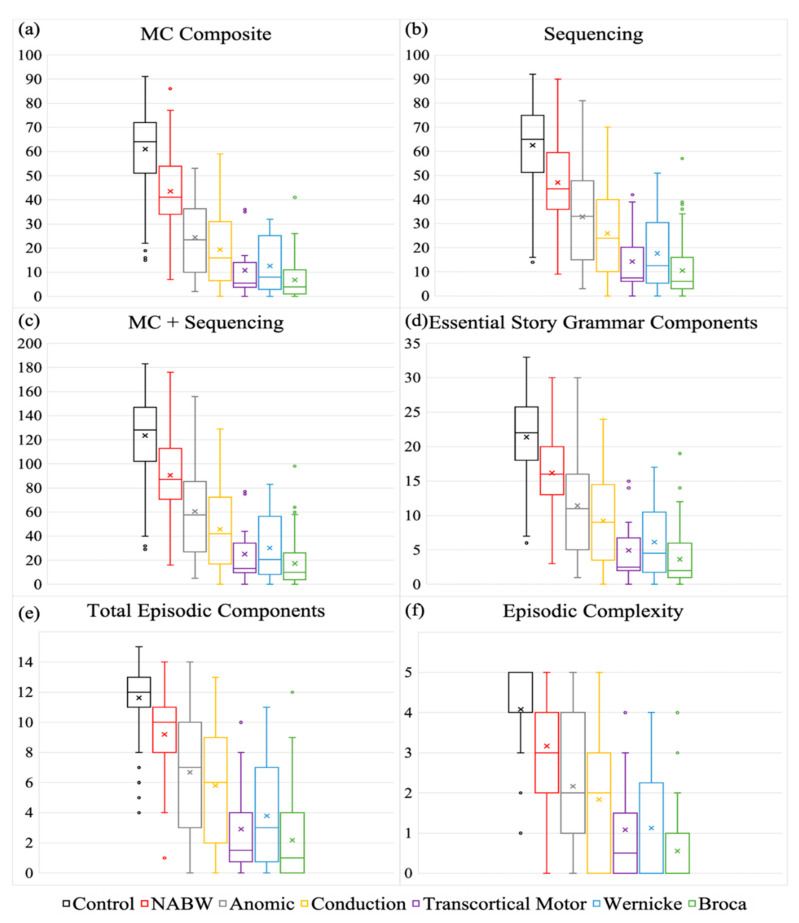
Boxplots of the distribution of scores for controls or persons not brain injured (PNBIs; **black**), persons not aphasic by WAB (NABW; **red**), persons with anomic aphasia (**gray**), persons with conduction aphasia (yellow), persons with transcortical motor aphasia (**purple**), persons with Wernicke’s aphasia (**blue**), and persons with Broca’s aphasia (**green**) for (**a**) main concept composite score; (**b**) sequencing score; (**c**) main concept composite + sequencing score; (**d**) essential story grammar components score; (**e**) total episodic components score; and (**f**) episodic complexity score. (In a boxplot, data are split into quartiles and the figure attributes are as follows: top of the box is the 75th percentile; bottom of the box is the 25th percentile; horizontal line within the box is the median of the dataset; × is the mean of the dataset; upper whisker is the line from top of the box to the upper bound, or maximum value; lower whisker is the line from the bottom of the box to the lower bound, or minimum value; circles are outliers).

**Figure 2 brainsci-11-00110-f002:**
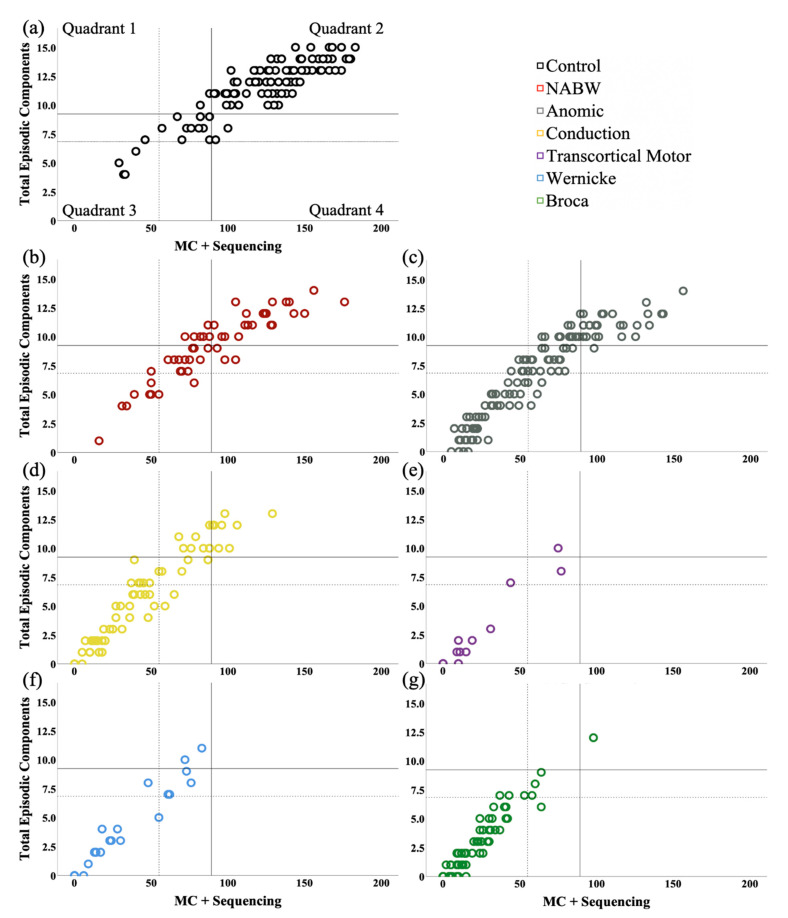
MSSG Classification for Cinderella narratives, plotted as a function of content and sequencing (MC+Sequencing; *x* axis) and overall episodic structure (total episodic components; *y* axis). Quadrants were defined by cutoff points at 1 *SD* (**solid line**) and 2 *SD* (**dotted lines**) below the mean for both variables. Quadrant 2 contains the highest scores for both dimensions, Quadrant 3 contains the lowest scores for both dimensions. Quadrant 1 contains low sequenced content and high overall episodic structure. Quadrant 4 contains high sequenced content and low overall episodic structure. (**a**) controls, or persons not brain injured (PNBIs) (**black**); (**b**) persons not aphasic by WAB (NABW) (**red**); (**c**) persons with anomic aphasia (**gray**); (**d**) persons with conduction aphasia (**yellow**); (**e**) persons with transcortical motor aphasia (**purple**); (**f**) persons with Wernicke’s aphasia (**blue**); (**g**) persons with Broca’s aphasia (**green**).

**Table 1 brainsci-11-00110-t001:** Demographic data reported for all PNBIs and PWAs, as well as by aphasia subtype.

	PNBI (*n* = 110)	PWA (*n* = 370)	Anomic (*n* = 122)	Broca’s (*n* = 85)	Conduction (*n* = 67)	NABW (*n* = 54)	Transcortical Motor (*n* = 12)	Wernicke’s (*n* = 24)
**Age (years) ^†^**	58.3 (±20.8) 20.0–89.5	62 (±12.6) 25–90.7	63 (±12.1) 32.7–85.7	58.3 (±12.9) 25.6–85.4	64.3 (±12.1) 30.9–90.7	60.7 (±13.7) 25–88	65 (±10.6) 47–83.6	67.3 (±10.8) 42.6–81.3
**WAB Aphasia Quotient ^††^**	N/A	72.8 (±19.8) 10.8–100	85.2 (±6.6) 63.4–93.5	51.2 (±15.1) 10.8–77.6	69.3 (±9.1) 47.8–90	96.6 (±1.8) 93.8–100	71.8 (±6.1) 59.8–80.3	52.2 (±14) 28.2–74.4
**Sex**	65 Female 45 Male	151 Female 219 Male	52 Female 70 Male	29 Female 56 Male	27 Female 40 Male	29 Female 25 Male	5 Female 7 Male	7 Female 17 Male
**Education (years) ^†††^**	15.8 (±2.5) 11–23	15.4 (±2.8) 7–25	15.7 (±2.8) 11–23	14.8 (±2.7) 8–23	15.2 (±3.1) 7–25	15.9 (±2.7) 12–21	14.1 (±2.3) 12–20	15.8 (±2.4) 12–20
**Race/Ethnicity**	92 Caucasian 3 African-American 3 Hispanic/Latino -- 12 Unknown	311 Caucasian 35 African-American 15 Hispanic/Latino 7 Other 2 Unknown	110 Caucasian 9 African-American 2 Hispanic/Latino -- 1 Unknown	65 Caucasian 12 African-American 5 Hispanic/Latino 3 Other --	58 Caucasian 5 African-American 1 Hispanic/Latino 3 Other --	44 Caucasian 2 African-American 7 Hispanic/Latino 1 Other --	9 Caucasian 3 African-American -- -- --	20 Caucasian 3 African-American -- -- 1 Unknown

PNBI: persons not brain injured; PWA: persons with aphasia; NABW: not aphasic by WAB. ^†^ 2 individuals (1 conduction, 1 Wernicke’s) are missing age data; ^††^ 5 individuals (2 anomic, 1 Broca’s, 1 NABW, 1 Wernicke’s) are missing WAB AQ data; ^†††^ 51 individuals (21 anomic, 11 Broca’s, 6 conduction, 10 NABW, 1 transcortical motor, 2 Wernicke’s) are missing education data.

**Table 2 brainsci-11-00110-t002:** Descriptive statistics for each MSSG variable, reported for all PNBIs and PWAs, as well as by aphasia subtype.

Statistic	Participant Groups							
	PNBI	PWA	ANO	BRO	CON	NABW	TCM	WER
MC Composite								
Mean	61.0	21.8	27.7	6.8	19.4	43.5	10.8	12.6
*SD*	17.1	18.8	17.5	7.7	14.6	16.6	12.5	11.9
Median	64	16	26	4	16	41	5.5	8
Range	15–91	0–86	2–75	0–41	0–59	7–86	0–36	0–32
Skew	−0.630	0.810	0.496	1.763	0.621	0.250	1.464	0.582
Kurtosis	0.209	−0.099	−0.468	4.064	−0.486	−0.076	0.992	−1.325
Sequencing								
Mean	62.5	26.5	32.8	10.5	25.9	47.1	14.3	17.6
*SD*	17.2	20.7	19.3	11.3	18.2	17.5	14.4	16.3
Median	65	24	33	6	24	44.5	7.5	12.5
Range	14–92	0–90	3–81	0–57	0–70	9–90	0–42	0–51
Skew	−0.706	0.563	0.284	1.544	0.407	0.202	1.121	0.688
Kurtosis	0.263	−0.605	−0.800	2.806	−0.924	−0.257	0.024	−0.837
MC Composite + Sequencing								
Mean	123.5	48.3	60.5	17.3	45.8	90.6	25.1	30.3
*SD*	34.2	39.3	36.7	18.9	32.7	34.0	26.8	28.1
Median	128	41	57.5	10	42	87	13	20.5
Range	29–183	0–176	5–156	0–98	0–129	16–176	0–77	0–83
Skew	−0.680	0.668	0.378	1.630	0.455	0.226	1.276	0.628
Kurtosis	0.252	−0.390	−0.654	3.341	−0.818	−0.172	0.444	−1.096
Essential Story Grammar Components								
Mean	21.4	9.2	11.4	3.6	9.2	16.2	4.9	6.1
*SD*	5.9	7.1	6.7	3.8	6.2	5.8	5.1	5.6
Median	22	8	11	2	9	16	2.5	4.5
Range	6–33	0–30	1–30	0–19	0–24	3–30	0–15	0–17
Skew	−0.591	0.537	0.321	1.468	0.343	0.160	1.215	0.664
Kurtosis	0.203	−0.617	−0.642	2.468	−0.914	−0.232	0.283	−0.871
Total Episodic Components								
Mean	11.6	5.5	6.7	2.2	5.8	9.2	2.9	3.8
*SD*	2.4	4.2	3.9	2.6	3.9	2.8	3.4	3.5
Median	12	5	7	1	6	10	1.5	3
Range	4–15	0–14	0–14	0–12	0–13	1–14	0–10	0–11
Skew	−1.088	0.198	−0.205	1.375	0.168	−0.611	1.216	0.655
Kurtosis	1.056	−1.314	−1.257	1.706	−1.198	0.099	0.107	−0.803
Episodic Complexity								
Mean	4.1	1.7	2.2	0.6	1.8	3.2	1.1	1.1
*SD*	1.0	1.7	1.6	0.9	1.7	1.3	1.4	1.5
Median	4	1	2	0	2	3	0.5	0
Range	1–5	0–5	0–5	0–4	0–5	0–5	0–4	0–4
Skew	−1.293	0.397	−0.049	1.633	0.451	−0.631	1.134	0.875
Kurtosis	1.520	−1.216	−1.308	2.253	−0.990	−0.240	−0.126	−0.806

PNBI: persons not brain injured, PWA: persons with aphasia, ANO: anomic, BRO: Broca’s, CON: conduction, NABW: not aphasic by WAB, TCM: transcortical motor, and WER: Wernicke’s.

**Table 3 brainsci-11-00110-t003:** Percent overlap of PNBIs with each aphasia subtype (top) and percent overlap of each aphasia subtype with PNBIs (bottom) for all MSSG variables.

	MC Composite	Sequencing	MC+Sequencing	Essential SG Components	Total Episodic Components	Episodic Complexity
PNBI overlap anomic	81%	87%	85%	96%	95%	100%
PNBI overlap Broca’s	13%	33%	20%	34%	56%	60%
PNBI overlap conduction	39%	64%	52%	70%	78%	100%
PNBI overlap NABW	96%	98%	96%	96%	95%	100%
PNBI overlap transcortical motor	8%	14%	9%	16%	24%	60%
PNBI overlap Wernicke’s	5%	25%	13%	25%	40%	60%
Anomic overlap PNBI	70%	76%	74%	74%	74%	76%
Broca’s overlap PNBI	14%	31%	25%%	27%	26%	34%
Conduction overlap PNBI	57%	67%	63%	67%	64%	69%
NABW overlap PNBI	98%	98%	98%	98%	98%	96%
Transcortical motor overlap PNBI	25%	33%	33%	33%	25%	50%
Wernicke’s overlap PNBI	33%	46%	38%	42%	42%	46%

**Table 4 brainsci-11-00110-t004:** Model distribution, Wald χ^2^ test statistic, *p*-value, and beta weight for all significant predictors of MSSG variable scores in the generalized linear models.

	PNBI Predictors		PWA Predictors	
MC Composite	Tweedie+Identity link		Gamma+Identity link	
	Age	χ^2^ = 24.8 *p* < 0.001 β = −0.387	Education	χ^2^ = 7.5 *p* = 0.006 β = 1.0
	Gender	χ^2^ = 10.0 *p* = 0.002 β_Female_ = 10.1	Gender	χ^2^ = 14.8 *p* < 0.001 β_Female_ = 9.2
Sequencing	Tweedie+Identity link		Gamma+Identity link	
	Age	χ^2^ = 20.0 *p* < 0.001 β = −0.355	Education	χ^2^ = 6.6 *p* = 0.10 β = 1.0
	Gender	χ^2^ = 10.266 *p* = 0.001 β_Female_ = 10.4	Gender	χ^2^ = 11.7 *p* = 0.001 β_Female_ = 8.9
MC+Sequencing	Tweedie+Identity link		Gamma+Identity link	
	Age	χ^2^ = 22.4 *p* < 0.001 β = −0.742	Education	χ^2^ = 7.0 *p* = 0.008 β = 2.0
	Gender	χ^2^ = 10.2 *p* = 0.001 β_Female_ = 20.5	Gender	χ^2^ = 13.6 *p* < 0.001 β_Female_ = 18.3
Essential Story Grammar Components	Poisson+Identity link		Gamma+Identity link	
	Age	χ^2^ = 29.6 *p* < 0.001 β = −0.118	Education	χ^2^ = 7.2 *p* = 0.007 β = 0.4
	Gender	χ^2^ = 14.6 *p* < 0.001 β_Female_ = 3.5	Gender	χ^2^ = 13.1 *p* < 0.001 β_Female_ = 3.2
Total Episodic Components	Poisson+Identity link		Tweedie+Identity link	
	Age	χ^2^ = 10.2 *p* = 0.001 β = −0.051	Education	χ^2^ = 6.4 *p* = 0.012 β = 0.2
			Gender	χ^2^ = 10.9 *p* = 0.001 β_Female_ = 1.9
Episodic Complexity	Multinomial+Cumulative logit link			
	Age	χ^2^ = 16.0 *p* < 0.001 β = −0.038	Education	χ^2^ = 7.2 *p* = 0.007 β = 0.1
	Gender	χ^2^ = 7.2 *p* = 0.007 β_Female_ = 1.1	Gender	χ^2^ = 20.3 *p* < 0.001 β_Female_ = 0.9

## Data Availability

A portion of the data analyzed in this study can be found here: https://aphasia.talkbank.org/. New data collected, analyzed, and created in this study will be shared with AphasiaBank and will be available to researchers, educators, and clinicians working with persons with aphasia according to the ground rules of the consortium.

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
