# Peer review of "Main Concept, Sequencing, and Story Grammar Analyses of Cinderella Narratives in a Large Sample of Persons with Aphasia"

_brainsci, 2021, doi:10.3390/brainsci11010110_

Round 1

Reviewer 1 Report

This is a very detailed analysis of main concept sequencing and study grammar (Main Concept, Sequencing, and Story Grammar, MSSG) in analyzing productions of the Cinderella story by persons with aphasia and non-brain injured controls. The results are impressive in distinguishing the two groups, even with relatively uncommon types of aphasia and those whose deficits are not picked up by the Western Aphasia Battery.

I would like to see the authors spend less time in the introduction in literature review, and more on the practical utility of the methodology, how easily it can be used, and how it would improve on current methods of aphasia diagnosis.

Author Response

Reviewer 1

I would like to see the authors spend less time in the introduction in literature review, and more on the practical utility of the methodology, how easily it can be used, and how it would improve on current methods of aphasia diagnosis.

Thank you for this thought-provoking critique. We respond here by highlighting information included in the original manuscript and by describing minor changes/additions for this revision in response to this critique.

We did not shorten the introduction, as we feel it is important to provide readers with the background information in the literature review, especially the more expanded sections (compared to MCA) on story grammar, sequencing, and combo concept + story grammar approaches for several reasons. First, while MCA is a single and well-described analysis, many different methods of analyzing story grammar have been utilized, and it is important for readers to understand where MSSG is situated in that landscape. Second, story grammar has not been as thoroughly or consistently studied in PWAs as it has been in persons with other brain injury (i.e., TBI) and communication disorders (i.e., cognitive-communication disorder), so readers of this manuscript will likely need the background to fully appreciate the analytic methods as well as the findings. Third, we sought to give credit where credit is due – MSSG is certainly not the first time that concepts or informativeness have been examined alongside story grammar or sequencing, and while we certainly deserve credit for this current combination and operationalization, we honor and recognize those attempts and approaches that preceded this work. 

The practical utility of MCA, which is a core component of MSSG analyses, is already described in 1.1. The practical utility of MSSG analyses as a package is yet unknown, and the focus of this manuscript is to complete the important work of establishing normative and comparative data. In the original manuscript, we highlighted practical utility in the results (e.g., practical significance reporting, percent overlap data, MSSG classification) and expanded on this in the discussion. We addressed ease of use in the first part of section 4 (~ lines 786-791), and in this revision we have added a sentence for more emphasis there. We focused much attention on diagnosis in section 4.1. Spotlight on PWAs-NABW.  We listed the tools needed to complete MSSG analyses and interpretation in 5. Conclusions, and in this revision we have added a sentence in this section to expand on how the MSSG Categorization can be used clinically.   

Reviewer 2 Report

I would like to thank you for the opportunity to read this very interesting study. The study explores mainly macrostructural  properties of PWA's narration of the Cinderella story. The results highlight differences between healthy controls and PWA, as well as between groups on the basis of aphasia-type; also, the study explores the predictive value of sociodemographic factors in the patients' narrative performance. Especially, the results on the performance of persons not brain injured are highly significant since they reveal language deficits that would otherwise go unnoticed.

I recommend minor revisions.

The minor revisions refer to the following points:

1. Do the studies [23], [24], [28], [29] and [34] that the authors cite in the Introduction involve information about the type of the aphasia of the patients? If they do, this information should be included in the Introduction.

2. lines 297-304. The protocol of administering the Cinderella test should move to Materials.

3. I personally found extremelly interesting the 'Non-essential workers' category in 4.2.4. that falls outside the scope of the present study. Besides the explanations posed by DeDe and Salis, would the authors care to include their thoughts on this issue, i.e. what do they believe is/are the reason(s) that PWA include non-relevant information in their narratives, besides processing speed deficits?

Author Response

Reviewer 2

1. Do the studies [23], [24], [28], [29] and [34] that the authors cite in the Introduction involve information about the type of the aphasia of the patients? If they do, this information should be included in the Introduction.

  1. Ulatowska, H.K., North, A.J., & Macaluso-Haynes, S. (1981). Production of narrative and procedural discourse in aphasia. Brain and Language, 13, 345-371.

The authors described their sample “at the time of experimental testing…three subjects were described as mild to moderate, while seven were only mildly aphasic…two patients were classified as having anterior lesions, four as posterior, and four as mixed”. In our manuscript, we described the participants from this study as “10 milder impaired PWAs” after referring to participants from both Ulatowska et al. studies as “PWAs with mild to moderate impairment”. In this revision, we have added parenthetical material regarding severity – “(7 mild, 3 mild-moderate)” – but we did not further speculate on subtype, given the vague neurological/lesion location information.

  1. Ulatowska, H.K., Freedman-Stern, R., Doyel, A.W., & Macaluso-Haynes, S. (1983). Production of narrative discourse in aphasia. Brain and Langauge, 19, 317-334.

The authors described their sample as “moderately aphasic”, with “8 patients classified as anterior, 5 as posterior, and 2 as mixed.” In our original manuscript, we described the participants from this study as “15 moderately impaired PWAs” and we have not made changes to this description in this revision.

  1. Bottenberg, D., Lemme, M., & Hedberg, N. (1985). Analysis of oral narratives of normal and aphasic adults. Clinical Aphasiology15, 241-247.

The authors described their sample as “Nine subjects were mildly impaired and one subject was moderately impaired”. No other descriptors re: categorization/subtypes were included. In our original manuscript, we described the PWAs as “(9 mild, 1 moderate)” and we have not made changes to this description in this revision.

  1. Lemme, M. L., Hedberg, N. L., & Bottenberg, D. E. (1984). Cohesion in narratives of aphasic adults. In Clinical aphasiology: Proceedings of the conference 1984 (pp. 215-222). BRK Publishers.

The authors used the same sample as [28]. No changes were made to this revision re: [28] or [29].

  1. Whitworth, A., Leitao, S., Cartwright, J., Webster, J., Hankey, G. J., Zach, J., ... & Wolz, V. (2015). NARNIA: a new twist to an old tale. A pilot RCT to evaluate a multilevel approach to improving discourse in aphasia. Aphasiology29(11), 1345-1382.

The authors described their sample as “Fourteen participants with mild-moderate aphasia” and specified in Table 2 the number that were mild or moderate. They included detailed assessment for a number of measures, but did not include further information re: subtype. We described the sample in the original manuscript version as “14 PWAs (7 mild, 7 moderate)” and we have not made changes to this description in this revision.

2. lines 297-304. The protocol of administering the Cinderella test should move to Materials.

This information has been moved to a new section, 2.2. Transcripts. This led to several other minor changes involving renumbering of later sections and renumbering of references 45 and 46.

3. I personally found extremelly interesting the 'Non-essential workers' category in 4.2.4. that falls outside the scope of the present study. Besides the explanations posed by DeDe and Salis, would the authors care to include their thoughts on this issue, i.e. what do they believe is/are the reason(s) that PWA include non-relevant information in their narratives, besides processing speed deficits?

We appreciate your interest, and we feel that study of processing speed deficits will be important for optimizing aphasia treatment. We have added a sentence stating this to the end of 4.1. Spotlight on PWAs-NABW. We have also expanded the 4.2.4. Non-essential Workers discussion and have added our current interpretation given the heterogeneity of this population.